# Tackling Infectious Diseases with Rapid Molecular Diagnosis and Innovative Prevention

**Rabeea F. Omar** *, **Maurice Boissinot** , **Ann Huletsky and Michel G. Bergeron**

Centre de Recherche en Infectiologie de l'Université Laval, Axe Maladies Infectieuses et Immunitaires, Centre de Recherche du CHU de Québec-Université Laval, Québec City, QC G1V 4G2, Canada; maurice.boissinot@crchudequebec.ulaval.ca (M.B.); ann.huletsky@crchudequebec.ulaval.ca (A.H.); michel.g.bergeron@crchudequebec.ulaval.ca (M.G.B.)
* Correspondence: rabeea.omar@crchudequebec.ulaval.ca

**Abstract:** Infectious diseases (IDs) are a leading cause of death. The diversity and adaptability of microbes represent a continuing risk to health. Combining vision with passion, our transdisciplinary medical research team has been focussing its work on the better management of infectious diseases for saving human lives over the past five decades through medical discoveries and innovations that helped change the practice of medicine. The team used a multiple-faceted and integrated approach to control infectious diseases through fundamental discoveries and by developing innovative prevention tools and rapid molecular diagnostic tests to fulfill the various unmet needs of patients and health professionals in the field of ID. In this article, as objectives, we put in context two main research areas of ID management: innovative infection prevention that is woman-controlled, and the rapid molecular diagnosis of infection and resistance. We also explain how our transdisciplinary approach encompassing specialists from diverse fields ranging from biology to engineering was instrumental in achieving success. Furthermore, we discuss our vision of the future for translational research to better tackle IDs.

**Keywords:** infectious diseases; prevention; diagnosis; antimicrobial resistance; innovation; technology transfer

## 1. Infectious Diseases Global Annual Mortality

According to the World Health Organization (WHO), worldwide, the top three leading causes of deaths are cardiovascular diseases, cancer, and infectious diseases (IDs) [1]. Global annual deaths from IDs have been declining steadily over the past 20 years from 208 deaths per 100,000 people in 2000 (12.7 million deaths in a total world population of 6.1 billion people at the time) down to 106 deaths per 100,000 people in 2019 (8.2 million deaths in a total world population of 7.7 billion people at the time) [2]. The main reasons for this reduction include affordable therapies and preventive measures.

## 2. Deaths Due to Resistance to Antibiotics/Antimicrobials

Antimicrobial (antibacterial, antiviral, antifungal, and antiparasitic) resistance (AMR) of infections is a growing global threat causing significant mortality. An estimated 4.95 million deaths were associated with AMR in 2019 [3]. The World Economic Forum predicts that global mortality due to antimicrobial resistance could reach 10 million people worldwide (4.73 M in Asia, 4.15 in Africa, 0.39 M in Latin America, 0.39 M in Europe, 0.317 M in North America, and 0.022 M in Oceania) [4]. The World Health Organization (WHO) is calling for a One Health approach to fight AMR and this is articulated around the appropriate use of antimicrobials [5].

## 3. Multifaceted Transdisciplinary Approach to Tackle Infectious Diseases

From the beginning, our research group envisioned a multifaceted approach (prevention, diagnostics, and therapeutics) to tackle and ultimately improve the management of IDs. We built a transdisciplinary team where an array of specialists focused on common tasks and exchanged knowledge on a weekly basis during the course of a series of parallel research projects each spanning 3–7 years. This was possible by combining several funding sources, involving multiple public and private organizations covering different sectors, ranging from fundamental and clinical research to applied engineering, industrialization, business models, and legal partnership frameworks. We believe that patient-oriented applied medical research providing data within a clinically actionable timeframe is the future to better serve unmet patients' needs. Patients should be in control of prevention; giving women control of their protection against sexually transmitted infections (STIs)/Human Immunodeficiency Virus (HIV) is crucial. Our team have been devoting their medical research to closing the loop, from identifying patients' problems at bedside, to finding solutions in the laboratory through product development and clinical testing, to finally bridging the gap of unmet needs at the patients' bedside. We also believe that in the future, researchers should be trained on the basics of business management and technology transfer. Licensing revenues help offset the decline in research support by funding government agencies, philanthropies, and the pharmaceutical industry.

## 4. New Vision That Is Changing ID Diagnosis Medical Practice

Since the mid-1990s, our team have been working to revolutionize the diagnosis of infectious diseases, which until then would have required at least two days because it relied on traditional microbial culture techniques dating from the time of Louis Pasteur. The first molecular diagnostic tests used in clinical practice were probe-based assays without nucleic acid amplification, but still required microbial culture [6]. The invention of the polymerase chain reaction (PCR) in 1985 [7] and its practical application through the use of thermostable polymerase [8], followed by the availability of PCR thermocyclers, paved the way for major changes in the culture-free molecular diagnosis of infectious diseases due to its high sensitivity and specificity, and ability to produce billion-fold copies of nucleic acid from a small number of microorganisms present in a sample. However, the widespread use of PCR in clinical microbiology laboratories has been made feasible by the introduction of closed-tube assays based on real-time PCR (rtPCR) detection using fluorescent technologies and new rapid PCR cycle instruments [9–12]. This eliminated the need for complex multi-step post-PCR analysis while reducing hands-on time and the risk of contamination by carryover amplification of previously generated amplicons [13,14]. This also resulted from a better understanding of the rules for designing efficient primers and probes and optimal PCR components [13,15,16], and the development of pre-PCR processing to extract microbial nucleic acid and remove PCR inhibitors in clinical samples [15,17,18]. By combining rtPCR and multiplex rtPCR molecular detection technology with the then emerging microbial genomics and rapid sample preparation process, we demonstrated that it was possible to detect microbes and their antibiotic resistance genes directly from clinical specimens in approximately one hour. Bringing these technological innovations into medical practice, Dr. Michel G. Bergeron created the company Infectio Diagnostic Inc. (IDI) (Quebec, QC, Canada) in 1995 and developed with his team the very first rtPCR test to receive approval from the Food and Drug Administration (FDA) in USA for detecting Group B *Streptococcus* directly from clinical specimens in less than one hour [IDI-Strep B™ (2002—FDA)] [19], a major breakthrough for the rapid diagnosis of infectious diseases. This was followed by another FDA-cleared test for detecting methicillin-resistant *Staphylococcus aureus* (MRSA) [IDI-MRSA™ (2004—FDA)] [20,21]. Furthermore, in collaboration with Becton Dickinson (BD), which acquired IDI in 2006, our group developed more rtPCR tests such as vancomycin resistance [BD GeneOhm™ VanR (2011—FDA)] and enteric pathogens on the fully automated BD MAX system [BD MAX™ Enteric Bacterial Panel (2014—FDA)]. Several tests have been developed on this automated system or adapted to this system such

as the MRSA assay. rtPCR tests are now used all over the world for the diagnosis and rapid detection of infections. The clinical impact of rapid rtPCR tests has been evaluated in the years since they were introduced, and they have been shown to reduce hospital-acquired infections, save lives, and reduce health care cost. For example, MRSA transmission rates while using standard MRSA culture (3 days) was 13.9 per 1000 patient days versus 4.9 while using IDI-MRSA$^{TM}$ (same day). This in turn allowed for an economy of USD 243,750 per 1000 patient days [22]. Rapid rtPCR tests have proven to be a powerful tool not only for the diagnosis of infectious diseases, but also for the prevention and control of infections, reducing the spread of antimicrobial-resistant pathogens.

The last two decades following the introduction of the first rtPCR tests in the clinic have seen the development of several commercial and laboratory-developed rtPCR and other molecular tests for the detection of a wide range of microorganisms and antibiotic resistances [23,24]. The list of nucleic acid-based tests (close to 400 tests) that have been cleared or approved by the US-FDA is available [25]. The automation of molecular tests has been an important step towards their wider use in clinical microbiology laboratories. Automated systems for nucleic acid extraction and purification were first introduced to reduce the labor associated with the manual sample preparation steps prior to molecular assay, making the recovery of nucleic acids more reproducible and reducing the risk of the cross contamination of samples [15]. Then, fully automated systems were developed that carry out all the steps from sample preparation to nucleic acid amplification and detection, saving labor time, making them easier to use, and, for some systems, enabling multiple samples to be analyzed at once to achieve the high throughput required in clinical microbiology laboratories for certain types of samples [26]. A new innovative approach was introduced in 2011 which includes commercial molecular assays that simultaneously detect and identify the multiple pathogens and resistance genes associated with a clinical syndrome in a single test such as bloodstream, respiratory, gastrointestinal, and central nervous system infections [27–29]. These syndromic panels have been shown to improve antimicrobial therapy and patient outcomes through better clinical decision-making, optimized laboratory workflow, and enhanced antimicrobial stewardship, but their effective use must be based on guidelines for implementation and interpretation [27–30].

Nucleic acid amplification technologies (NAATs) other than PCR have also been developed, the most widespread being a variety of isothermal amplification (IA) methods that do not require temperature cycling, thus providing a simpler and less costly procedure for the rapid detection of nucleic acids from clinical samples. Some IA-based tests are now incorporated into a variety of commercial molecular tests [23,31–34]. However, an important advantage of rtPCR fluorescence detection over other molecular technologies is its ability to detect microorganisms quantitatively or semi-quantitatively, based on a standard curve [15]. Knowledge of microbial load can be used to infer a patient's response to treatment, or to provide information to distinguish infection from colonization [35]. Digital PCR (dPCR), the latest generation of PCR, has emerged as a promising new PCR technology which provides absolute quantification without the need for a standard curve. dPCR involves partitioning the PCR solution into a large number of droplets, and the reaction is carried out in each partition individually, with endpoint fluorescence detection and Poisson statistics for the absolute quantification of nucleic acid targets. It has been shown to be more tolerant to PCR inhibitors and more sensitive and reproducible compared to rtPCR [36]. Different commercial dPCR systems are available which are based on chamber/chip-based dPCR (cdPCR) platforms and droplet-based dPCR (ddPCR) platforms, and at least three dPCR SARS-CoV-2 tests have been granted emergency use authorization (EUA) approval [36]. Several studies have shown the potential of dPCR for the detection and clinical management of infections [36–39]. However, the use of dPCR in clinical settings is limited due to the high cost of instruments and consumables compared to rtPCR, as well as the complexity of the workflows which require greater hands-on time [36]. Continuous improvement aimed at reducing complexity and costs should enable the wider use of this powerful technology in the future.

Molecular diagnostic technologies were later taken to a new height, bringing diagnosis to the point of care (POC) as portable, simple, and fast tests, an effective way of further reducing the turnaround time (TAT) for rapid molecular tests, which depends on the diagnostic cycle, i.e., the number and duration of steps in the diagnostic process required from prescribing the test, collecting, and transporting the clinical sample to the laboratory, to transmitting the result to the physician [40–42]. With this goal in mind, Dr. Bergeron established GenePOC Inc. (Quebec, QC, Canada) in 2007 (acquired by Meridian Bioscience Inc. (Cincinnati, OH, USA) in 2019). The transdisciplinary R&D team assembled experts in chemistry, instrumentation, microbiology, medical device manufacturing, microfluidics, molecular biology, nanotechnology, medicine, optics, physics, and regulatory affairs, whose work has been integrated by engineers. This translated into designing and building state-of-the-art Revogene®, Meridian Bioscience, Quebec City, QC, Canada (Figure 1), a fully automated molecular platform integrating clinical sample processing with single and multiplex testing capabilities while requiring minimal hands-on time [43]. Revogene® automates sample homogenization, dilution, cell lysis, the conversion of RNA templates into DNA using reverse transcription, nucleic acid amplification, and the detection of the amplified PCR products [42]. User intervention is only required to transfer the clinical sample into the disposable cartridge (PIE) and insert the PIEs into the Revogene® carousel rotor. When compared to similar POC technologies, the Revogene system has a small space requirement, a less expensive instrument (~USD 30,000), a similar price per test (~USD 28), and easy waste management [44].

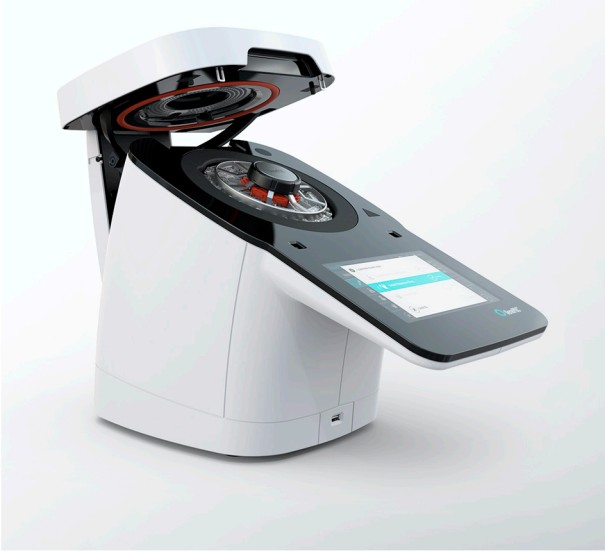

**Figure 1.** Revogene®—Automated molecular testing platform.

Several rtPCR diagnostic tests have been developed on the Revogene® platform and approved by the US-FDA and European Medicines Agency (EMA), such as Instrument Revogene® (2017—FDA, CE (Conformité Européenne)), GenePOC™ GBS LB (2017—FDA, CE), GenePOC GBS DS (2017—CE), GenePOC Cdiff (2017—FDA, CE), GenePOC Strep A (2019—FDA, CE), GenePOC Carba C (2019—FDA, CE), and Revogene® SARS-CoV-2 (2022—FDA EUA (emergency use authorization)).

In 2015, four nucleic acid-POC tests based on PCR amplification and isothermal amplification received for the first time a Clinical Laboratory Improvement Amendment Certificate (CLIA) of Waiver, Compliance, or Accreditation status in the USA. CLIA-waived tests are defined as "simple tests that have an insignificant risk of an erroneous result" [45], and thus can be performed by less skilled personnel [42,46]. Today, there are more than thirty nucleic acid tests for the detection of single or multiple microorganisms (syndromic panels) which have received CLIA-waiver, including twenty-five SARS-CoV-2 tests un-

der FDA EUA authorization [46–48]. The CLIA-waiver represents a major step towards opening-up molecular diagnostic tests close to patients for use in decentralized (laboratory) settings in hospitals such as close to the emergency department (ED), intensive care unit, and delivery room, or outside of hospitals such as in outpatient clinics, physicians' offices, pharmacy laboratories, and remote area facilities (dispensaries of developing countries) [42]. For example, the use of a CLIA-waived POC test for SARS-CoV-2 in an ED has recently been shown to accelerate clinical decision-making, improve patient management, and significantly reduce the length of stay in the ED for patients requiring outpatient care [49].

The COVID-19 pandemic triggered remarkable advancements in molecular diagnostics, leading not only to the widespread laboratory adoption of rtPCR and other molecular innovative testing technologies coming from both existing and new manufacturers without previously available tests, but also to a greater public awareness of the importance of rapid molecular test performance, such as high sensitivity, high specificity, and rapid TAT, in the management and control of infectious and highly transmissible pathogens [50,51]. More than 260 molecular SARS-CoV-2 tests have been granted FDA EUA approval [47]. Molecular tests have now become an integrated component of guideline-recommended practices for a variety of infections, being accepted as the standard of care, replacing conventional methods, and routinely applied to detect the presence of these infection-causing pathogens, [23]. For example, rapid nucleic acid tests have replaced culture for the detection of respiratory virus infections in the majority of virology laboratories, which have chosen to abandon viral culture because of its lower sensitivity, long TAT and hands-on time, and the need for technical expertise [52]. However, the cost of molecular tests can be an obstacle to their implementation in some clinical settings, particularly in developing countries. However, demonstrating the health care cost-effectiveness and the availability of affordable molecular POC tests, with their simplicity, short TAT, and wide accessibility, could help overcome this obstacle, especially in resources-limited settings [53]. Furthermore, in developing countries, price can be subsidized by contributions from richer countries and international organizations. By offering more powerful tools for the earlier and more accurate detection of infectious diseases, molecular diagnostics represents a veritable revolution whose considerable clinical impact is constantly being demonstrated for several infections, with advantages including a more appropriate use of pathogen-target therapies, a reduced length of hospital or emergency department stay, decreased unnecessary antibiotic use, improved infection control practices and antimicrobial stewardship, and reduced overall health care costs [30].

## 5. Innovative Woman-Controlled Infection Prevention

Women need effective, affordable, and accessible means under their control to protect themselves because of their increased risk of contracting sexually transmitted infections compared to men due to their anatomy. Women have large vaginal and cervical mucosal surface that is exposed to STIs including HIV during unprotected vaginal intercourse [54]. Therefore, it is imperative to give women control over their own protection. In addition, women, like men, should have universal protection against STIs including HIV and unintended pregnancy. Although progress has been made in the area of women's reproductive and sexual health since the first female contraceptive appeared in the early 1960s, unfortunately there still are unpleasant side effects for many women today [55]. Women need better choices, mostly non-hormonal on-demand contraceptives, protection against STIs, and more control over their reproductive and sexual health [56,57]. However, the prevention of STIs remains primarily in the hands of men, as the male condom is still the most effective way of reducing the risk of all STIs including HIV. Women, particularly those living in countries where gender parity is low, often have little control over condom use by their male sexual partners [58]. In addition, men have easy access to protection against STIs/HIV and unintended pregnancy via the male condom, yet women need an appointment, a doctor's visit, and a prescription to have a vaginal product to give the same protection. Now, it is time for the US-FDA and other regulatory authorities to recon-

sider making such crucial protective woman-controlled products freely available over the counter (OTC), or behind the pharmacy counter to ease access for women in need. Despite some progress made, women's empowerment is being threatened to be set back in many countries, even some developed ones [57,58]. Research on women's health has long been and remains a neglected field [58,59]. Apart from the female condom, which is expensive and cumbersome to use compared to the male condom, there are no protective means under the control of women [60]. In addition, male condoms are only used in one-third of risky sexual intercourse acts, putting women at risk of infection [61]. Over the past two decades, researchers have been working on various tactics to protect women against STIs/HIV. They have tested various antimicrobial polymers and formulations such as vaginal microbicides to protect women against STIs/HIV. Other researchers have focused on using antiretrovirals in polymer gel formulations to protect against HIV only. We have previously discussed in detail these different tactics of prevention and the many tested products [60].

On International Women's Day, 8th of March 2005, the then Democratic Senator Barack Obama (who later became the US 44th Democratic President from 2009 to 2017) introduced (with Democratic Senator Jon Corzine and Republican Senator Olymbia Snowe) the "Microbicide Development Act". This act was a cornerstone legislation that would encourage scientific leadership on this issue and strengthen research and development programs at the National Institutes of Health (NIH), Centers for Disease Control (CDC), and USAID. 'Vaginal' microbicides are a class of products under development that women could use to protect themselves from contracting HIV and other STIs. Senator Obama published an article on that day entitled "A new hope for preventing the spread of HIV" [Obama, B. A new hope for preventing the spread of HIV. Chicago Defender 8 March 2005 Vol. XCIX, No. 215]. It is worth mentioning that mathematical modeling predicted that over 3 years, 2.5 million infections could be averted if a microbicide that is 60% effective against HIV were used by only 20% of women in half of all sexual acts that do not involve a condom [62]. Sexually active women need protection against STIs as well as against unintended pregnancy. Therefore, dual protection woman-controlled products now contain microbicides and spermicides and are referred to as multipurpose prevention technology. It is crucial for these new woman-controlled products to be accessible and affordable. In our opinion, the price of a single use (on-demand) vaginal applicator (pre-filled) should be less than USD 5 for each protected vaginal intercourse. Being accessible and affordable will encourage more and more women from all social and economic settings to use such protective products that empower them.

The Invisible Condom® vaginal gel developed by our research team concerned with women's health and gender parity is one of the tools that will enable women to empower their sexual and reproductive health [63]. For the intravaginal delivery of the gel, we realized that the available conventional vaginal applicators equipped with a single apical hole would not give the required maximal mucosal coverage for the optimal protection of women against HIV/STIs. Therefore, in collaboration with industrial engineers, we designed a unique vaginal applicator equipped with multiple apical and lateral holes for maximal and homogenous vaginal/cervical mucosal coverage (Figure 2).

The Invisible Condom® woman-controlled vaginal gel is designed to protect against STIs including HIV, HSV-2, gonorrhoea, chlamydia, trichomonas, and candida, as demonstrated in the laboratory [64,65]. The gel offers both a physical barrier (poloxamer gel polymer alone) and a chemical barrier (poloxamer gel plus microbicide/spermicide-sodium lauryl sulfate (SLS)). Both poloxamer 407 NF (Spectrum Chemical, Gardena, CA, USA) and SLS NF (Spectrum Chemical) are common ingredients in household items that we safely use on a daily basis such as tooth paste, dental rinse, shampoo, nasal decongestant, kids bubble bath, shower gel, etc. The gel formulations are compatible with male condoms. Furthermore, the gel is imperceptible (not perceived by male sexual partners) as it is hydro-soluble and mixes with natural vaginal secretions. The unique gel formulation is thermoreversible and has rheological characteristics that ensure good mucosal adhesion. In fact, we have

shown in a magnetic resonance imaging (MRI) clinical trial that the gel persisted in the vagina during simulated sex (with an artificial phallus and 30 vaginal thrusts) and stayed in place between 4 to 8 h to offer protection immediately upon application, during, and after sexual intercourse [66]. Moreover, our novel vaginal applicator performance was compared with six other conventional applicators available on the Canadian market, and we demonstrated that our uniquely designed vaginal applicator homogenously delivers the gel formulation over the vaginal/cervical mucosa of women, giving a uniform distribution and maximum coverage for optimal protection.

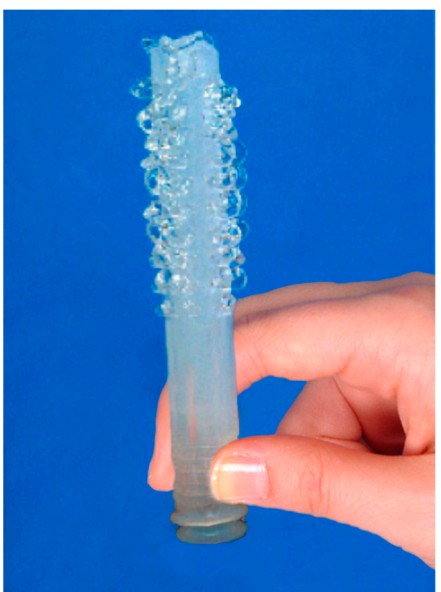

**Figure 2.** Invisible Condom® unique vaginal applicator.

Of importance, in completed Phase I and Phase II clinical trials in ~500 women from two different populations (Canadian and African women) who had about ~30,000 vaginal gel applications, we have demonstrated the safety and tolerability of the Invisible Condom® and its special vaginal applicator. Moreover, the vaginal pH and the normal vaginal microbiota, as determined by Nugent score, the presence of lactobacilli, and the production of $H_2O_2$ by vaginal lactobacilli were not affected by the gel alone or gel plus microbicide [67,68]. A phase III anti-STIs/HIV efficacy clinical trial has been designed and we are presently seeking financing.

The Invisible Condom® also protects against unintended pregnancy. The gel-SLS formulation completely inhibited human sperm motility, and was also shown to be contraceptive in a rabbit model [69]. More recently, we conducted a Pre-Phase III (Phase IIB) trial to assess the spermicidal and contraceptive efficacy of the Invisible Condom® vaginal gel in Canadian women of childbearing age and their stable male sexual partners. For in vitro spermicidal effect (30 men), 98% of sperms were immotile in the presence of gel-SLS. For post-coital test (the survival of human sperm in cervical mucous after penile vaginal intercourse with ejaculation inside the vagina; 30 couples), 99% of sperms were immotile in the presence of gel-SLS. Finally, for contraceptive efficacy, among 30 couples, overall, 410 vaginal intercourses in 95 menstrual cycles were protected (during a 3-month period of gel-SLS use before each vaginal intercourse with a probability of 24 conceptions prevented according to Wilcox's table) [70]. The Invisible Condom® vaginal gel with its demonstrated spermicidal and contraceptive effect may present a potential non-hormonal contraceptive option for women, especially those with contraindications to use or those who do not want to use hormonal contraceptives. A larger Phase III contraceptive efficacy trial is warranted.

Furthermore, despite research advances and available knowledge, we are still learning about vaginal microbiota. Evidence of this is the fact that during one of our clinical trials,

we discovered a new bacterium that was isolated from the vaginal sample of a woman diagnosed with bacterial vaginosis [71], and was named Criibacterium bergeronii after Prof. Bergeron and our ID research center, CRI. The new isolated bacterium represents a new species and the first species of a new genus. It is not clear if it has a pathologic role. It was later shown by other researchers to be found in the vagina of women with gynecologic cancer [72], and to be associated in large numbers with idiopathic cutaneous ulcers in children treated with azithromycin [73].

## 6. Vision for the Future in Managing Infectious Diseases

While NAATs are now occupying a growing place in clinical microbiology laboratories, these methods can only detect known pathogens and their already characterized targeted genes. As target gene variants evolve, new key genes appear, and previously unknown pathogens emerge, novel molecular testing strategies are bound to spawn. Metagenomics, now facilitated by next-generation sequencing (NGS) methods, permits the agnostic detection of any microbial gene present in a clinical sample with a sensitivity similar to that of specific rtPCR assays [74]. NGS can identify microbes, their subtypes, their virulence factors, and their AMR genes [75]. Sample preparation methods have been designed to enrich the detection of microbes, and cell free circulating microbial nucleic acid analysis by a commercial NGS service is now available [76]. These approaches can improve the diagnosis of infections and contribute to the surveillance, monitoring, control, and prevention of IDs in humans, animals, food, and the environment, thus aiming for optimal One Health interventions. In addition to research for developing, simplifying, and validating the pipelines required for generating and analyzing NGS data, collaborative efforts are necessary to establish national and international guidelines, governance principles, clinical and cost utility demonstrations, secure information technology, and legal/ethical frameworks [77]. Major obstacles to implementing genomics' technologies still remain, especially in developing countries, for which special pricing structures and capacity building for disseminating know-how and expertise are essential for global health. As stated by the WHO science council, the potential of genomics will not be fully achieved unless these technologies can be globally deployed and sustainably supported in all countries [78]. Numerous challenges remain to be solved by collaboration between developed and developing countries for NGS to be routinely implemented into microbiology and public health laboratories.

The future of clinical microbiology will integrate genomics, metagenomics, resistomics, proteomics, transcriptomics, metabolomics, glycomics, and artificial intelligence (AI) to analyse multiomics data and develop innovative strategies for patient-controlled prevention, rapid diagnosis, and appropriate treatment, allowing a better management of infectious diseases.

Translating R&D into clinical practice requires transdisciplinary work where researchers from various fields tackle common problems together in a collaborative way long enough for a learning exchange to take place [79,80]. Once this occurs, real-value-added innovations are shaped by engineering processes to create game-changing opportunities.

To ensure a better future, researchers need to maintain and enforce the established conditions of success to address emerging global challenges and future pandemics. They need to continue being innovative, agile, proactive, and inclusive, and to have more transdisciplinary and international collaborations. We posit that innovative transdisciplinary teams should be driven by curiosity, audacity, openness, and simplicity. Passionate by curiosity, team members must be audacious in their common endeavour, while being open to other disciplines, persons, and aim for the simplest possible way to address even the most daunting ID challenges.

**Author Contributions:** R.F.O., Writing—original draft, Conceptualization, Investigation, Validation, and Writing—review and editing. M.B., Conceptualization, Validation, and Writing—review and editing. A.H., Validation and Writing—review and editing. M.G.B., Strategic discussions. All authors have read and agreed to the published version of the manuscript.

**Funding:** This research received no external funding.

**Acknowledgments:** This article is dedicated to all CRI researchers who have been working hard over many years to save lives and improve the quality of life of patients. It is also dedicated to the patients participating in clinical trials to make new prevention tools, new diagnostic technologies, and new treatments available to patients around the world. The authors wish to thank Jean-Luc Simard for his help.

**Conflicts of Interest:** M.G.B. is the founder of two university start-ups: Infectio Diagnostic Inc. (IDI) and GenePOC Inc. These two companies were acquired by the multinationals BD Diagnostics and Meridian Biosciences. M.G.B., A.H., and M.B. received financial compensation for those transactions at the time and are not holding any participation nor are expected to receive any more compensation. M.G.B., A.H., and M.G.B. are co-inventors on some rapid molecular diagnostic technologies. R.F.O. and M.G.B. are co-inventors of the woman-controlled prevention technology.

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
