# Peer review of "Tackling Infectious Diseases with Rapid Molecular Diagnosis and Innovative Prevention"

_2036-7449, doi:10.3390/idr16020017_

Round 1

Reviewer 1 Report

Comments and Suggestions for Authors

1. After section 2, I recommend section devoted to prevention of infectious diseases. This short section should describe various tactics of prevention such as antimicrobial polymer coatings. After such text, section 3 in which you describe your contribution to the field will be perfect.

2. In section 5 please write what type of poloxamer was used and its producer. In the reference 63 there is no name of poloxamer that was used only in few experiments.

Author Response

  1. After section 2, I recommend section devoted to prevention of infectious diseases. This short section should describe various tactics of prevention such as antimicrobial polymer coatings. After such text, section 3 in which you describe your contribution to the field will be perfect.

We agree with the reviewer on the need for adding a short section on various tactics of prevention for STIs. However, we don’t think it should be placed after section 2 since the background for rapid diagnostics and prevention are within their respective sections 4 and 5. Therefore, we have added a short paragraph (lines 245-253) about prevention (in section 5 rather than after section 2) just before describing our contribution to the field, which makes better links.

Thank you for your suggesting that improved this section.

  1. In section 5 please write what type of poloxamer was used and its producer. In the reference 63 there is no name of poloxamer that was used only in few experiments.

It’s Poloxamer 407 NF (Spectrum); this is now precised in section 5 of the revised article.

Reference 63 was also replaced by another more relevant reference: Roy et al. Thermoreversible gel formulations containing Sodium Lauryl Sulfate or n-Lauroylsarcosine as potential topical microbicides against sexually transmitted diseases. Antimicrob. Agents Chemother. 2001, 1671-1681.

Thank you

Reviewer 2 Report

Comments and Suggestions for Authors

The work of Omar and cols. is a narrative review regarding the molecular diagnosis and innovative prevention of infectious diseases. In general, the manuscript is adequately structured and well written, but the work is not informative.

Author Response

The work of Omar and cols. is a narrative review regarding the molecular diagnosis and innovative prevention of infectious diseases. In general, the manuscript is adequately structured and well written, but the work is not informative.

This review gives important details about 2 essential topics of tackling infectious diseases: rapid molecular diagnostic and the new vaginal multipurpose prevention technologies under control of women. We also explain how our transdisciplinarity approach was instrumental to achieve success.

Thank you

Reviewer 3 Report

Comments and Suggestions for Authors

In this manuscript a review of new ways of approaching the diagnosis and prevention of infectious diseases is made.

The following comments are made:

1. Line 53. Write what means “STIs/HIV”. Before putting an abbreviation, put the meaning. Correct throughout the text.

2. Line 160. Report and discuss “Revogene” costs.

3. Line 192. Put what TAT means.

4. In section “4. New Vision That Is Changing ID Diagnosis Medical Practice”, Discuss what the costs of this technology are and what happens when the Health System does not have the money to access this technology.

5. Line 274. Put what MIR means

6. Lines 286 and 306. “vaginal flora”, that term is no longer used, is vaginal microbiota.

7. In section “5. Innovative Women-Controlled Infection Prevention”. Discuss costs, availability and how to encourage use of the product.

8. In section 6. Discuss the costs, since if money is needed to do PCR, this is greater if NGS is done, and the latter requires experts in bioinformatics to make the interpretations. Discuss both things in health systems of developing or underdeveloped countries.

Author Response

In this manuscript a review of new ways of approaching the diagnosis and prevention of infectious diseases is made.

The following comments are made:

  1. Line 53. Write what means “STIs/HIV”. Before putting an abbreviation, put the meaning. Correct throughout the text.

Agreed, and is now defined in the revised article.

  1. Line 160. Report and discuss “Revogene” costs.

We added text in the revised article (lines 169-171) to explain Revogene system cost.

  1. Line 192. Put what TAT means.

Already previously defined on original line 151 (first time), now line 155.

  1. In section “4. New Vision That Is Changing ID Diagnosis Medical Practice”, Discuss what the costs of this technology are and what happens when the Health System does not have the money to access this technology.

We added text (lines 208 to 214) to discuss the cost of this technology and its potential impact in resources-limited settings.

  1. Line 274. Put what MRI means

Now defined in the revised article.

  1. Lines 286 and 306. “vaginal flora”, that term is no longer used, is vaginal microbiota.

Changed in revised article.

  1. In section “5. Innovative Women-Controlled Infection Prevention”. Discuss costs, availability and how to encourage use of the product.

Text discussing that (lines 269-274) is now added in revised article.

  1. In section 6. Discuss the costs, since if money is needed to do PCR, this is greater if NGS is done, and the latter requires experts in bioinformatics to make the interpretations. Discuss both things in health systems of developing or underdeveloped countries.

In the revised article, we addressed this issue and the need for global collaboration for NGS implementation (lines 379-386).

Thank you.

Reviewer 4 Report

Comments and Suggestions for Authors

The authors present a manuscript entitled "Tackling Infectious Diseases with Rapid Molecular Diagnosis and Innovative Prevention."    The topic is highly intriguing and the content is well-prepared.    The authors have compiled valuable information.    However, the objective of this work remains unclear.     It is not mentioned anywhere, leaving the manuscript seemingly directionless.     The objectives should be explicitly stated at the beginning of the text.     Consequently, the conclusion appears somewhat disconnected.

I believe the manuscript to be original because it assesses the latest molecular methodologies in infectious disease diagnosis. It would serve as valuable citation material, as well as an excellent educational resource.

Furthermore, the sections seem disjointed and lack cohesion.  This aspect also requires improvement.

The authors conclude, succinctly, that for a better future, researchers must uphold conditions of success, be innovative, agile, proactive, and collaborative.  They should inspire the next generation with passion, promoting the application of knowledge and vision.   Transdisciplinary teams should be motivated by Curiosity, Audacity, Openness, and Simplicity, seeking simple solutions to complex challenges.  While I find this conclusion reasonable, as I mentioned earlier and suggested to the authors, the objectives are not clear.  Therefore, the conclusion is also unclear.

Improving upon these points, I would be pleased to accept this manuscript for publication as an article.

Author Response

The authors present a manuscript entitled "Tackling Infectious Diseases with Rapid Molecular Diagnosis and Innovative Prevention.   The topic is highly intriguing and the content is well-prepared.  The authors have compiled valuable information.   However, the objective of this work remains unclear.   It is not mentioned anywhere, leaving the manuscript seemingly directionless.   The objectives should be explicitly stated at the beginning of the text.   Consequently, the conclusion appears somewhat disconnected.

The objectives are now explicitly stated and clearly defined at the beginning of the article (at the end of the abstract).

I believe the manuscript to be original because it assesses the latest molecular methodologies in infectious disease diagnosis. It would serve as valuable citation material, as well as an excellent educational resource.

Furthermore, the sections seem disjointed and lack cohesion.  This aspect also requires improvement.

In the revised article, we made sure that the sections link well with cohesion and the text flows well. For example, we have removed some parts on prevention of pregnancy when they were not directly linked to STI prevention.

The authors conclude, succinctly, that for a better future, researchers must uphold conditions of success, be innovative, agile, proactive, and collaborative.  They should inspire the next generation with passion, promoting the application of knowledge and vision.   Transdisciplinary teams should be motivated by Curiosity, Audacity, Openness, and Simplicity, seeking simple solutions to complex challenges.  While I find this conclusion reasonable, as I mentioned earlier and suggested to the authors, the objectives are not clear.  Therefore, the conclusion is also unclear.

The revised conclusion is now shorter, clearer and better linked with the objectives now stated in the abstract.

Thank you.

Improving upon these points, I would be pleased to accept this manuscript for publication as an article.

Reviewer 5 Report

Comments and Suggestions for Authors

The authors present a manuscript (review) entitled "Tackling Infectious Diseases with Rapid Molecular Diagnosis and Innovative Prevention", which covers a very important topic. They describe briefly the old, well-known methods, and some that a considered novel. The authors even add their own experience. 

The manuscript is interesting, easy to read and understand. I would like to propose some minor changes:

1. Line 34: "important mortality" - In my opinion, this word combination does not sound good and sort of inappropriate. May be you can use "high mortality" or "significant mortality" instead.

2. Line 38: WHO is already given in full in line 25. 

3. Line 45: "ID" is not presented in full the first time it is mentioned.

4. Line 53: STIs/HIV - Please, give them in full. 

Author Response

The authors present a manuscript (review) entitled "Tackling Infectious Diseases with Rapid Molecular Diagnosis and Innovative Prevention", which covers a very important topic. They describe briefly the old, well-known methods, and some that considered novel. The authors even add their own experience. 

The manuscript is interesting, easy to read and understand. I would like to propose some minor changes:

  1. Line 34: "important mortality" - In my opinion, this word combination does not sound good and sort of inappropriate. May be you can use "high mortality" or "significant mortality" instead.

Agreed, and changed it to ‘significant mortality’ in the revised article.

  1. Line 38: WHO is already given in full in line 25. 

Agreed, and now revised.

  1. Line 45: "ID" is not presented in full the first time it is mentioned.

Now defined in the revised article on line 27 (first time).

  1. Line 53: STIs/HIV - Please, give them in full. 

Defined in the revised article.

Thank you